



MAGNETIC RESONANCE
Open Access Discussions

# Second harmonic electron paramagnetic resonance spectroscopy and imaging reveal metallic lithium depositions in Li-ion batteries

Charles-E Dutoit[1,2], Hania Ahouari[1,3], Quentin Denoyelle[4], Simon Pondaven[2,5], and Hervé Vezin[1,2]

[1] Université Lille Nord de France, CNRS, UMR8516, LASIRE, 59655 Villeneuve d'Ascq, France

[2] Centre de Résonance Magnétique Electronique pour les Matériaux et l'Energie, 59655 Villeneuve d'Ascq, France

[3] Université de Lille, FR2638-IMEC-Institut Michel-Eugène Chevreul, 59655 Villeneuve d'Ascq, France

[4] SAFT, Corporate Research, 111 Boulevard Alfred Dancy, 33074 Bordeaux, France

[5] TotalEnergies OneTech R&D, Centre de Recherche de Solaize (CRES), chemin du canal, BP 22, 69360 Solaize, France

**Correspondence:** Charles-E Dutoit (charles.dutoit@univ-lille.fr)

**Abstract.** We have investigated the metallic lithium particles nucleation following lithiation and delithiation steps of the graphite electrode using X-band electron paramagnetic resonance (EPR). Metallic lithium aggregates like dendrites and/or filaments which are formed during electrochemical cycling on the graphite anode are complex structures which may lead to internal short-circuit and safety issues. Understanding and following, in real conditions, this nucleation process is necessary to improve the development of Li-ion batteries. The complexity to detect metallic lithium structures inside Li-ion batteries depends on the number of EPR lines and their linewidth. The presence of lithiated graphite phases affects the detection of micrometric Li-metal elements. Herein, we report a new approach using cw-EPR spectroscopy and imaging combining the first and the second harmonic detection schemes to provide evidence of the metallic lithium aggregates nucleation in these negative electrodes. Although the first harmonic gives all the EPR signals present in the sample, it is found that the second harmonic EPR signal is mainly sensitive to metallic lithium depositions.

## 1 Introduction

The family of rechargeable Li-ion batteries (LIBs) is known to be used in a wide variety of applications, from portable electronics to electric vehicles, due to their high specific capacities, good lifespan and their decreasing cost (Tarascon and Armand (2001); Armand and Tarascon (2008)). More particularly, with their high capacities of 372 mAh.g$^{-1}$, graphite electrode materials are the most widely used as anodes in such batteries. Due to the low voltage open circuit potential of graphite, metallic lithium is susceptible to deposit onto the graphite particles at the negative electrode during the charge of the cell, i.e. during the graphite lithiation. This parasitic reaction is more likely to happen at low temperature, high charge rates and high state-of-charge. In normal operating conditions, the lithium plating level is low, but participates to cell capacity loss due to lithium consumption in the metallic aggregates and in the formation of the additional solid electrolyte interphase (SEI). For more severe charge conditions, lithium can even form dendrites, which can lead to safety issues in case of internal short-circuits. The non-uniform metallic lithium plating on the graphite anode is the main limitation for a faster charge protocol, essential for the development of transport electrifications (Liu et al. (2018); Waldmann et al. (2018); Weiss et al. (2021)). Following in



real time and operating conditions this degradation process is challenging, yet necessary to keep improving the Li-ion battery performance (Foroozan et al. (2020); Finegan et al. (2020)).

This type of information requires the use of non-destructive methods which leave the sample intact during and after measurements, without destroying sub-micrometric metallic aggregates newly formed. Magnetic resonance techniques appear to be suitable for ex-situ, in situ and operando measurements of Li-metal structures due to their low frequency fields using radio/microwave frequency for nuclear magnetic resonance (NMR) and electron paramagnetic resonance (EPR) respectively, which penetrate samples with negligible energies.

NMR spectroscopy and imaging are well established techniques to investigate redox processes in Li-based electrochemical batteries but also to detect metallic particles which can grow during charge and discharge (Bhattacharyya et al. (2010); Chandrashekar et al. (2012); Fang et al. (2022)). EPR, which is the electronic equivalent of NMR, is most convenient to probe in depth Li-metal depositions like bulks, dendrites or metallic filaments through its high sensitivity to conduction electrons (Sathiya et al. (2015); Wandt et al. (2015, 2018); Niemoller et al. (2018); Nguyen et al. (2020); Dutoit et al. (2021)), compared to NMR spectroscopy. However, in the case of graphite (de)lithiation, the resolution of the Li-metal EPR spectrum is limited by the presence of broad lithiated graphite signals at a resonance field near the one of Li-metal. As a consequence, the Li-metal signal is overlapped inside the lithiated graphite spectrum and, sometimes, is not recognizable.

In this work, we report the direct observation of metallic lithium depositions with micrometric sizes on the negative graphite electrode using, for the first time, the second harmonic detection mode. We correlate the first and the second harmonic detection schemes of EPR spectroscopy and imaging to obtain information about the nucleation and the spatial distribution of metallic lithium structures in LiFePO$_4$/graphite batteries.

## 2 Experimental details

### 2.1 Electrochemical cell:

In this study, a LiFePO$_4$ (LFP)/Graphite cell was considered using a LiPF$_6$ lithium salt dissolved in carbonate solvents. After usual formation cycles, cycling of the cell was carried out at 20 °C with a regime of C/2 performed on 90% of the state-of-charge (SOC) of the cell. Aged cell was cycled until 30% of capacity loss. The cell was then discharged and dismantled in an argon-filled glovebox. Graphite electrode was washed three times using dimethyl carbonate (DMC) and dried. Anode was sampled by cutting out rectangles taken from the center of the anode and then placed in a sealed tube before EPR analysis.

### 2.2 Electron paramagnetic resonance:

Continuous wave (cw) electron paramagnetic resonance measurements were carried out at room temperature using a conventional X-band Bruker E500 spectrometer operating at around 9.6 GHz. The microwave power into the cavity was set to 0.2 mW in order to avoid saturation of the EPR signal. The 100 kHz modulation depth of the magnetic field was chosen as 0.2 mT or less to prevent distortion of the apparent EPR spectrum due to over-modulation. Conversion time and time constant were set





to 40.94 ms and 20.48 ms respectively. Simulation of cw-EPR spectra was done using EasySpin package for Matlab (Stoll and Schweiger (2006)). The first harmonic spectrum was fitted using the sum of two phase shifted Lorentzian functions defined in eq. (1). The asymmetric ratio A/B of the EPR line was obtained considering the first derivative EPR spectrum for lithiated graphite signal and the second derivative EPR spectrum for the metallic lithium aggregates.

Spatial-spatial and spectral-spatial images were collected using a field-of-view of 20 mm and a gradient strength of 175 G/cm with a size of 512 x 512 pixels resulting in a pixel size of 39.1 $\mu$m. The high resolution spatial-spatial images were recorded at room temperature using a deconvolution of the acquired projections under a magnetic field gradient from a signal recorded without gradient. Finally, EPR images were filtered with a back-projection. 330 projections were recorded in the spectral-spatial images with a spectral resolution of 1024 points and a pixel size similar to the one of spatial-spatial images. A filtered back-projection of the acquired projections was performed to get high resolution images for signals with a peak-to-peak linewidth lower than 10 G.

## 3    Results and Discussion

The reversible lithiation and delithiation of the graphite anode during a C/2 charge rate is analyzed at room temperature to provide evidence or not of metallic lithium traces in our LiFePO$_4$/graphite battery. Galvanostatic discharge curves are shown in Figure 1a, for the first and the last cycle before opening. As expected for a LFP/graphite cell, the potential is nearly constant during all the discharge, due to the flat open circuit potentials of LPF and graphite. At the beginning of life, the graphite plateaus are well visible, but almost disappear at the end of life. Indeed, for these cells the main ageing phenomenon is the solid electrolyte interphase (SEI) build-up, coming with loss of active lithium and graphite polarization increase. Figure 1b shows a representative X-band cw-EPR signal of the graphite anode recorded before electrochemical cycling (black line). The spectrum exhibits a single and broad dysonian-shaped EPR line with a g-factor of about 2.01 (resonance field of 341.4 mT at 9.61 GHz) and a peak-to-peak linewidth $\Delta B_{pp} \sim 3$ mT. It is well known that the g-factor of a radical is characterized by a specific environment (similar to the chemical shift in NMR spectroscopy). Typically, a g-factor of 2.01 may be attributed to an oxygen centered radical. Such an EPR signal is classically found in graphite electrode materials (Wang et al. (2021); Grey et al. (2023)). As expected, initially the metallic lithium EPR signal is featureless, consistent with pure graphite materials without Li-metal impurities.

An example of the ex-situ EPR spectrum of the graphite anode after electrochemical cycling is given in Figure 1b (red line). The shape of the EPR signal displays a different general pattern compared to the pristine spectrum with a line centered at a value g $\sim$ 2.0036, smaller than the one observed in the pristine state. Furthermore, the EPR line appears narrower than the pure graphite signal. Also this spectrum is characterized by a dysonian-shaped line more intense than in the pristine sample considering the same mass of matter. It can be seen that this signal has a peak-to-peak linewidth of about 0.3 mT and an asymmetric ratio A/B $\sim$ 2, as expected for the lithiated graphite (Wandt et al. (2018)). Indeed, the cycled cell was discharged before opening, but it corresponds to a state where the graphite is still partially lithiated.





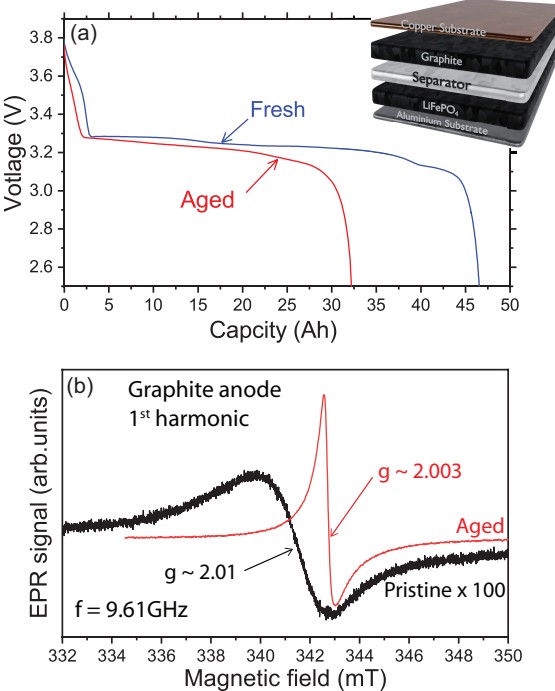

**Figure 1.** X-band cw-EPR spectroscopy of the graphite anode. (a) Galvanostatic cycling profile of the LiFePO$_4$/graphite cell with a schematic representation of the electrode materials stacking. The blue curve (fresh) is the first discharge and the red curve (aged) is the last discharge of the cell. (b) EPR spectra of the pristine (black) and aged (red) electrodes using the 1$^{st}$ harmonic detection scheme. Note that the black curve has been amplified (x100) to compare the signal intensities.

In Figure 2, we represent a simulation of the lithiated graphite signal considering two different cases. In the first one, a first derivative of a single dysonian function is used to simulate the EPR spectrum. As we can see, the main features of the experimental spectrum are not correctly reproduced suggesting at least another additional contribution hidden under the lithiated graphite spectrum. The ability to resolve this second contribution depends on the number of functions used in the

simulation. In the second case, the signal was simulated using a sum of two contributions : (i) a relative broad dysonian function with an asymmetric ratio A/B $\sim$ 1.6 and a linewidth $\Delta B \sim$ 1 mT for the lithiated graphite species Li$_x$C$_6$ ($0 <$ x $\leq$ 1); (ii) a narrow dysonian line with A/B $\sim$ 1.8 and $\Delta B \sim$ 0.2 mT. It is worth noting that this second EPR line is possibly over-modulated due to the modulation amplitude of 0.2mT used in this experiment. The modulation amplitude value of 0.2mT was chosen due to the apparent peak-to-peak linewidth of the spectrum observed before analysis (around 1mT). Consequently, the

real linewidth of this second signal is necessary smaller than 0.2mT. This additional EPR line is assigned to traces of metallic lithium aggregates with a size slightly bigger than the skin depth, here $\delta_{mw} \sim$ 1.1 $\mu$m at 9.6 GHz. Indeed, as discussed in a previous EPR investigation of symmetric Li-metal/Li-metal cells (Dutoit et al. (2021)), the EPR lineshape is influenced by the metal thickness compared to $\delta_{mw}$ due to the excitation of spins located exclusively inside the skin depth. When the metal




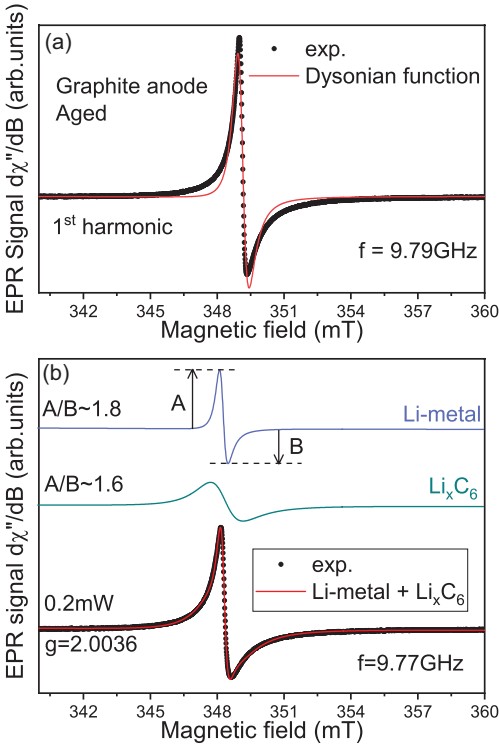

**Figure 2.** Identification of the second EPR contribution hidden under the lithiated graphite signal . (a) Simulation of the lithiated graphite anode aged signal recorded at room temperature using exclusively one dysonian function. (b) Same simulation using two dysonian contributions. The green line represents the lithiated graphite spectrum ($Li_xC_6$) and the blue line is indicative to Li-metal.

thickness is bigger than $\delta_{mw}$, a dysonian EPR lineshape is observed with an asymmetric ratio A/B $\gg$ 1 (Dyson (1955); Feher and Kip (1955)). In contrast, if the metal thickness is smaller than $\delta_{mw}$, a pure lorentzian line is obtained with an asymmetric ratio A/B = 1. It is important to note that the EPR spectrum of the graphite anode recorded after the first half lithiation (electrode potential $\sim$ 86 mV) does not show distinguishable Li-metal contribution (see supplementary Figure S1).

No rigorous theoretical models for porous metallic lithium micro-particles are available and we chose to estimate the smallest metallic lithium particle size from the empirical equation (Gourier et al. (1989)) defined by:

$$\frac{dP}{dB} = \alpha \frac{1 - \chi^2}{(1 + \chi^2)^2} - \beta \frac{2\chi}{(1 + \chi^2)^2} \tag{1}$$

with $\chi = \gamma(B - B_{res})T_2$, $\gamma$ representing the electron gyromagnetic ratio, $\alpha$ and $\beta$ being the dispersion and absorption parameters respectively. From this expression and the parameters $\alpha$ and $\beta$ characterizing the line shape and thus the asymmetric ratio A/B, we estimate a micrometric size of about 1.6 $\mu$m.

In summary, the cw-EPR signal of lithiated graphite samples, recorded at X-band, contains a mixture of two overlapping contributions, i.e., lithiated graphite complexes and traces of micrometric metallic lithium aggregates. In the absence of rigor-





ous simulations, such metallic signals could be indistinguishable. Furthermore, the difficulty in distinguishing such traces of Li-metal at low potential due to the weaker Li-metal line which is masked by the intense $Li_xC_6$ peak has been already reported in the literature (Wang et al. (2021)). To clearly observe such metallic structures in EPR measurements, we have to improve the spectral selectivity. We suggest here a new approach by playing with the harmonic detection schemes of the EPR spectrum

(Schwarz and Norbert (1980)). In a conventional X-band EPR spectrometer, there is an option which allows the detection of the second harmonic of the modulated EPR spectrum, i.e., the second derivative of the absorption signal (simultaneously with the first harmonic mode). The first harmonic mode (first derivative) is routinely used in standard EPR and gives EPR signals of all magnetic species present in the sample, here the lithiated graphite and the Li-metal signals. The second harmonic, which is mainly used for resolution enhancement of unresolved hyperfin structures, gives a better spectral resolution for overlapping

EPR signals. Indeed, the $n^{th}$ harmonic being sensitive to the slope of the EPR signal, broader spectrum tends to be less prominent than sharper peak with higher order harmonics without spectral distortion caused by a slightly over-modulation (Wilson (1963); Tseitlin et al. (2011); Yu et al. (2015)). Figure 3 shows an example of X-band spectra obtained after the galvanostatic cycling and recorded using the first (black) and the second (red) harmonic detection modes. As discussed previously, the first harmonic mainly reveals the presence of lithiated graphite (no Li-metal signal directly distinguishable). As we can see, the

second harmonic spectrum contains exclusively one contribution centered, in the limit of the X-band spectrometer resolution, at a similar measured resonance field than the lithiated graphite. Furthermore, this signal displays a very sharp dysonian EPR line consistent with the Li-metal signal. This interpretation is reinforced by the results presented in Figure 3b. As we can see, the pseudo-modulation (PM) of the simulated EPR signals, corresponding to the Li-metal and $Li_xC_6$ defined in Figure 2, displays two different behaviors. Although the signal of PM[$Li_xC_6$] shows a large and flattened line, the signal of PM[Li-metal]

is intense and in good agreement with the $2^{nd}$ harmonic line recorded experimentally. We tested the $2^{nd}$ harmonic spectrum assignment by using a symmetric cell with metallic lithium disks as both electrodes (see supplementary Figure S2). Initially, the $1^{st}$ harmonic EPR signal consists of a broad and dysonian lineshape characteristic of the bulk lithum signal. After the short circuit, the EPR spectrum exhibits two contributions: (i) a large and dysonian line corresponding to the Li-metal electrode (bulk) and (ii) a very sharp spectrum characteristic of sub-micrometric metallic Li particles. Finally, the $2^{nd}$ harmonic con-

tains only the Li-metal sub-micrometric information and shows a similar shape and g-factor to the one obtained in Figure 3a confirming our hypothesis. These results show that in the case of the graphite lithiation, the metallic aggregates formed during electrochemical cycling are better resolved in this detection mode.

Now, let us discuss the spatial distribution of these metallic particles. Figure 4 focuses on the ex-situ EPR images recorded on cycled samples. A gradient of 175 $G.cm^{-1}$ was used for spatially encoding both complexes (lithiated graphite and Li-

metal) with a high resolution due to their respective linewidth ($\leq 10$ G). Especially for metallic lithium element signals which display a peak-to-peak linewidth of about 1 G (Niemoller et al. (2018); Maresch et al. (1986)). In these examples, we used the spatial-spatial detection mode to get information about the location of aggregates in two perpendicular spatial directions **Y** and **Z**. Furthermore, we correlated the spatial-spatial images with spectral-spatial images to obtain spectroscopic information (lineshape, resonance field, asymmetric ratio A/B, . . . ). These spectroscopic parameters are crucial to clearly validate the nature

and the origin of each signal visible in EPR images (Dutoit et al. (2021)).





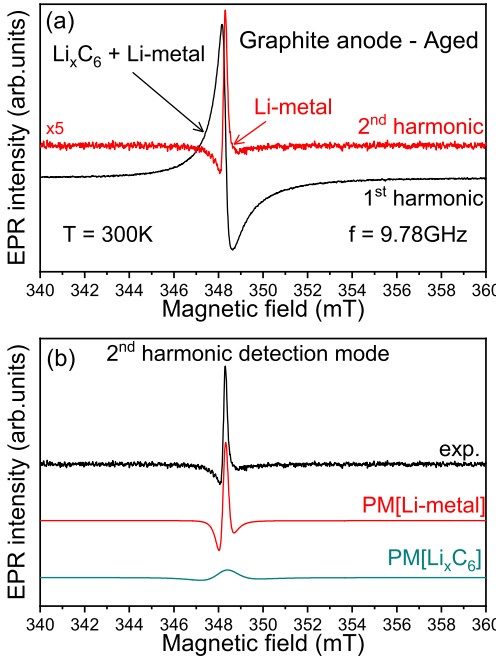

**Figure 3.** $2^{nd}$ harmonic X-band EPR detection scheme. (a) EPR spectra of lithiated graphite anode aged recorded at room temperature using the first (black) and the second (red) harmonic detection modes. (b) Pseudo-modulated (PM) EPR spectra recorded from the simulated signals Li-metal (PM[Li-metal]) and $Li_xC_6$ (PM[$Li_xC_6$]) defined in Figure 2.

Figure 4a-b show the EPR images recorded using the spatial-spatial and the spectral-spatial detection schemes respectively. The sample appears in the center from the intense EPR signals and its apparent shape is similar to the real shape with dimensions of about 25 mm x 2.5 mm. The spatial-spatial image confirms the non-homogeneous distribution of lithiated graphite species with spots mainly located on the bottom part of the sample. Some additional very intense signals (red) displaying a non-uniform distribution are clearly visible. This result suggests that some aggregates are more sensitive to the microwave field $\mathbf{b}_{mw}$. The corresponding spectral-spatial image confirms the presence of lithiated graphite species characterized by a relative broad dysonian EPR shape centered at a value g $\sim$ 2.0036 and displaying a peak-to-peak linewidth of about 0.3 mT (Figure 4b).

In order to distinguish and locate metallic lithium structures in the sample, we introduced here the correlation between the first and the second harmonic EPR images. This approach is new to the best of our knowledge. As shown before, in our electrochemical system, the second harmonic EPR spectrum is exclusively sensitive to the metallic lithium aggregates. Figure 4c shows the second harmonic spatial-spatial image of the same sample. Intense spots observed here seem to be correlated as those initially found in the standard detection mode. This result is the indication of Li-metal nucleation at the graphite anode surface and confirms that these aggregates are mainly located near the lithiated graphite regions. It is worth noting that the

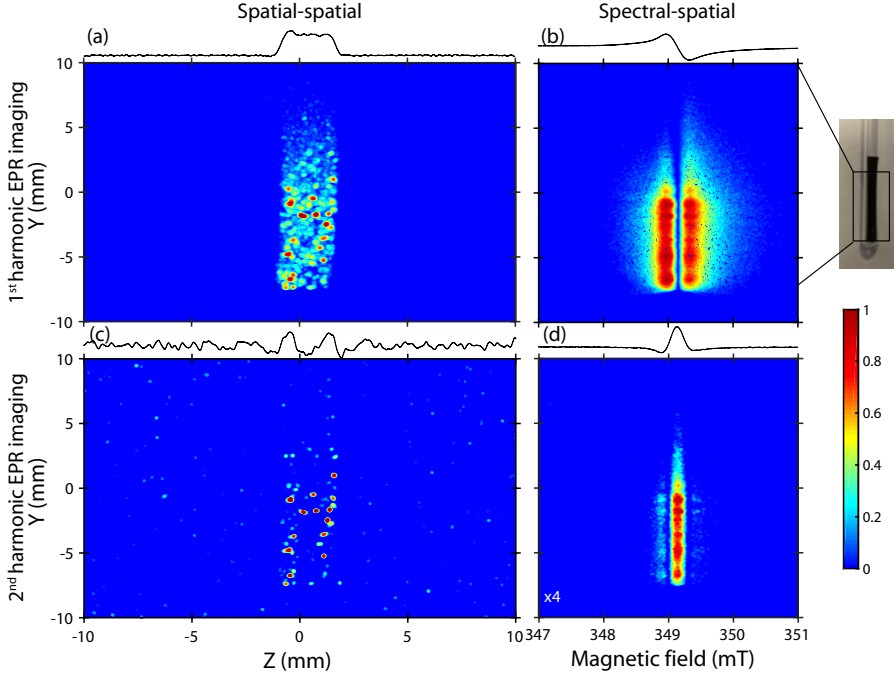

**Figure 4.** X-band EPR detection and location of Li-metal depositions. (a-b) Spatial-spatial and spectral-spatial images of the lithiated graphite anode recorded using the standard detection mode ($1^{st}$ harmonic). (c-d) $2^{nd}$ harmonic spatial-spatial and spectra-spatial images. For clarity, the spin contribution of the spectral-spatial image is indicated by the absolute value of the EPR spectrum where the red color represents the positive and negative lobes respectively. The color code is indicated by the color bar and illustrate the apparent amplitude of the signals.

pixel size used for recording the EPR images is around 39.1 $\mu$m which does not allow to clearly visualize the particles with a dimension close to 1.6 $\mu$m estimated by EPR spectroscopy.

## 4    Conclusions

To conclude, EPR spectroscopy is a nondestructive, rapid and sensitive method to detect micrometric and/or sub-micrometric Li-metal elements. The aim of this investigation was monitoring the metallic lithium aggregates nucleation on the graphite
anode following lithiation and delithiation steps using multi-mode EPR spectroscopy and imaging. It was shown that the second harmonic detection scheme is sensitive to the Li-metal structures with a size slighlty bigger than the skin depth. This effect allows to distinguish the spectroscopic signature of the metallic element when this one is overlapping in the lithiated graphite signal. We provide the correlation between the first and the second harmonic detection modes of EPR spectroscopy and EPR imaging to follow the Li-metal deposition. To date, and to the best of our knowledge, the second harmonic detection
mode was never used to clearly distinguish Li-metal plating/stripping in the graphite electrodes. This result offers an alternative approach for Li-based batteries paving the way for the detection and location of Li-metal aggregates.



*Code availability.* Matlab is a commercial software from MatlabWorks and the Easyspin package is available from https://easyspin.org/

*Data availability.* The dataset that support the findings of this investigation is available here: https://doi.org/10.5281/zenodo.10623150.

*Author contributions.* SP, QD and HV designed the project. CED performed EPR measurements, CED and HV interpreted the results The manuscript was written by CED through contributions of all authors. All authors given approval to the final version of the manuscript.

*Competing interests.* At least one of the (co-)authors is a guest member of the editorial board of Magnetic Resonance. The authors have no other competing interests to declare.

*Acknowledgements.* This work was supported by the Centre National de la Recherche Scientifique (CNRS) and by TotalEnergies Hybrid Storage program under a joint laboratory cr2me (Centre de Résonance Magnétique Electronique pour les Matériaux et l'Energie). The authors are grateful to SAFT for providing electrode materials, separator and electrolyte used in their batteries. We thank Bernard Simon for valuable discussions about the presence of metallic lithium structures in anodes.



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
