# Peer review of "Second harmonic electron paramagnetic resonance spectroscopy and imaging reveal metallic lithium depositions in Li-ion batteries"

_Magnetic Resonance, 2024_

## Author Response (AR1)

Point-by-point reply to reviewers and editor for the manuscript:

*Second harmonic electron paramagnetic resonance spectroscopy and imaging reveal metallic lithium depositions in Li-ion batteries*, by C-E. Dutoit, H. Ahouari, Q. Denoyelle, S. Pondaven and H. Vezin

The authors would like to thank the reviewers for their careful reading of this manuscript. Their comments and suggestions have, we hope, helped us to improve the quality of this article.

**Reviewer 1**

This paper is interesting.

We thank the reviewer for this positive appraisal of our work.

Comment #1: However, the main problem is that the modulation is set to 0.2 mT, which is much larger than the signal of lithium dendrite which is less than 0.05mT. Therefore, I suggest to repeat the experiments with the modulation of 0.05 mT, especially Figure 4.

**Response:** Thank you for this comment, which is absolutely right. Indeed, in the case of dendritic lithium structures, the EPR spectrum is expected very sharp (around 0.05mT and less) as reported in a variety of EPR investigations. But also, the EPR spectrum should exhibit a pure Lorentzian line shape unlike the micrometric lithium aggregates exhibit a Dysonian shape. This line shape effect is due to the skin depth ($\delta_{mw}$ = 1.1 µm for $Li^0$ at X-band) which limits the microwave penetration inside the conductor (i.e. through the metal thickness d). If $d>\delta_{mw}$ the EPR line exhibits an asymmetric peak (Dysonian with A/B > 1) whereas if $d<\delta_{mw}$, the EPR line is a symmetric and intense (Lorentzian with A/B=1).
Initially, we performed EPR measurements using an amplitude modulation of 0.2mT to follow the EPR signature of the lithiated graphite which gives a linewidth around 0.4mT. At this step of investigations, we did not expect to observe a metallic lithium EPR signature. But after analysis using two Dysonian functions, we performed the second harmonic investigation to confirm the presence and the origin of the other hidden contribution which is assimilated to micrometric lithium structures.
In our case, we did not detect dendritic structures and the reason is twofold. On one hand, no sudden drop to 0V characterizing a short-circuit was detected during electrochemical cycling. On the other hand, no symmetric EPR spectrum was observed as expected for sub-micrometric structures like dendrites.
We detected micrometric lithium particles with the size much higher than 1.5 µm which are much bigger than the skin depth. As a consequence, the linewidth of such particles is much larger than 0.05mT. Even if the EPR signal of the metal lithium aggregates, recorded using the first harmonic detection scheme is possibly over-modulated, the second harmonic mode does not show an excessive over-modulation for the EPR peak. In contrary, recorded a slightly over-modulated EPR line using the second harmonic mode increase the signal-to-noise ratio S/N.
As suggested by the reviewer, in the case of dendritic structures, an amplitude modulation less than 0.05mT will be necessary to get semi-quantitative information about such particles.

Comment #2: Thank you for the response. Since graphite is conductive, the Dysonian lineshape might comes from lithiated graphite rather than the dendrites.

**Response:** Thank you for this comment. Exactly, the EPR signature of the lithiated graphite (LixC6) exhibits a relatively large Dysonian line shape compared to the metallic lithium signal. In our study and on the aged sample, we detect such LixC6 spectrum characterized by an asymmetric ratio A/B around 1.6 and a linewidth of around 1mT. However, an additional signal is also detected and exhibits an asymmetric ratio A/B around 1.8 with a linewidth of around 0.2mT. This signal is possibly over-modulated (as explained in the previous comment) and its "real" linewidth is necessary smaller than 0.2mT as expected for Li-metal with micrometric size. It is worth noting here that we do not observe dendritic lithium structures which gives a symmetric line shape (A/B=1) but only micrometric lithium aggregates (much larger structures than dendrites) which give a Dysonian line shape (A/B>1). This last point is important because while micrometric lithium structures display a Dysonian line shape like LixC6, the difference between both materials comes from their respective linewidth. This is the reason why, using the second harmonic detection mode which is sensitive to the slope of the spectrum and hence sensitive to the sharpest line, we detect only the micrometric lithium contribution.

Comment #3: The explanation sounds OK. However, in my personal experiences, it is very difficult to observe metallic lithium depositions (which is not dendrites) in first cycle with C/2. That is why I doubt the results. It would be better if you show the photo of the electrode to see whether the metallic lithium depositions exist.

**Response:** This comment is absolutely right. As we showed in our supplementary (see figure S1) during the first cycle no spectroscopic signature of metallic lithium aggregates was observed. Indeed, we recorded the EPR spectrum of a graphite anode after the first half charge of the first cycle. This result is shown in Figure S1 and provide evidence of no additional signal from Li-metal but only the EPR spectrum of $Li_xC_6$ at this state of charge (SOC). However, in our EPR investigation, the aged sample, which presents an EPR spectrum from $Li_xC_6$ and an additional EPR spectrum from Li-metal, has been analyzed by EPR spectroscopy an imaging after undergoing more than 2000 (dis)charge cycles and not only 1. Even if the number of cycles is not clearly reported in our manuscript, this information is given by the sentence: Aged cell was cycled until 30% of capacity loss (experimental details). This last sample having undergone a large number of electrochemical cycles may present some degradation signature such as traces of metallic lithium aggregates.

We added few words clarifying the number of electrochemical cycles "Aged cell … several thousand times …" on page 2, line 46. We cannot show the photo of the electrode because this investigation was carried out several months ago and we no longer have the sample.

Comment #4: Thank you for the information. Now it is more or less OK. At last, for the end of this conversion, it would be useful if some photos were taken at that time.  Besides, thin dendrites might be as important to probe as the thick metallic deposition.

**Response:** Your comment is absolutely right. In the case of dendrite growth, a special attention is needed to understand the $Li^0$ nucleation process during cycling and avoid a serious risk of explosion. Such metallic structures will exhibit a pure Lorentzian lineshape (using the 1st harmonic mode) and the corresponding second harmonic spectrum will show a better signal-to-noise ratio (S/N) and a better apparent amplitude than the one expected for micrometric aggregates.

Comment #5: By the way, in the manuscript, there is no clear text stating that EPR is only detecting thick lithium deposits, not thin lithium deposits or lithium dendrites. This makes it easy for the reader to think that this article is detecting lithium dendrites, which are often more common. It was therefore suggested that the two different deposits should be clarified in the main text.

**Response:** Thank you for that remark which shows that we have not been clear on this point. In our study, we chose to use the term "micrometric size" to characterize the lithium aggregates rather than "sub-micrometric size" which, usually, characterize dendritic structures.

As suggested by the reviewer, we clarified this point on page 4, line 96 and on page 6, line 111 concerning the non-dendritic feature of these metallic lithium structures by the word non-dendritic.

**Reviewer 2**

The authors report a new cw-EPR approach to selectively detect Li metal signals by making use of the second harmonic detection. This method was demonstrated on a lithiated graphite electrode for Li-ion batteries. This new approach offers a certain degree of selectivity of the EPR detection towards Li metal, may find broad applications in understanding Li-ion batteries. This reviewer would like to recommend it for publication after minor revisions, as suggested below:

We thank the reviewer for his reading of the manuscript, which allows us to address some clarifications.

Comment #1: Since the major novelty of this work is the application of second harmonic detection to differentiate the Li metal signal from the $Li_xC_6$, the theory behind second harmonic detection and its selectivity towards narrow components should be (at least briefly) described. Furthermore, where is the boundary between being selective and non-selective in terms of EPR signal linewidth?

**Response:** Thank you for this comment. Metallic $Li^0$ species exhibit always a homogeneous EPR spectrum for which the EPR linewidth is mainly dominated by the spin-spin relaxation time $T_2$. It is possible to improve the spectral selectivity by playing with the relaxation (and then with the linewidth) of the detected signal. The second harmonic mode is sensitive to $T_2$, and usually a relatively broad signal (short $T_2$) gives an EPR signal using this detection mode. However, such signal appears flat in comparison to an EPR signal with a long $T_2$.
We added the sentence "i.e. sensitive to the spin-spin relaxation time $T_2$" and two bibliographic references have been added (on page 6, line 121-122, in red) concerning the selectivity of the second harmonic detection mode.

1) *Pali et al, J Magn Reson B. (1996), 113, 151-159*
2) ***Marsh et al, J. Chem. Soc., Perkin Trans. 2, (1997), 2545-2548***

**Comment #2:** Is it possible to distinguish dendritic and mossy Li structures by the second harmonic detection?

**Response:** Yes, it is possible to distinguish mossy and dendritic lithium structures using the second harmonic detection scheme. Indeed, the EPR spectrum of $Li^0$ is indicative to the variety of morphologies. In the case of thick $Li^0$, the EPR spectrum exhibits an asymmetric lineshape, also name Dysonian, with a linewidth around 0.15mT. In the case of mossy and dendritic lithium structures, the EPR signal is slightly Dysonian and pure Lorentizan respectively. The second harmonic EPR spectrum of mossy will be slightly asymmetric whereas the corresponding EPR signal of dendritic $Li^0$ will be symmetric. Furthermore, mossy $Li^0$ deposited during cycling exhibits a linewidth around 0.03mT whereas dendritic $Li^0$ shows a linewidth around 0.005mT as reported by Niemöller *et al* (Sci. Rep. **8**,1–7 (2018)). Consequently, dendritic $Li^0$ will be much more amplify with the second harmonic mode than the mossy $Li^0$ signal.

Furthermore, the new ELEXSYS II integrate today the new features for Signal Processing Unit (SPU) with Digital lock-in with up to 5 simultaneously detected harmonics with simultaneous detection of 0 and 90-degree modulation phases. In the case of multi-component with a similar g-factor, we believe that using multi-harmonic detection mode will be an invaluable method to clearly distinguish each contribution.

**Comment #3:** How the simulation is set up in EasySpin should be described in detail.

**Response:** We only simulated the EPR signal from a theoretical model using the sum of two Lorentzian phase shifted functions defined in equation (1) in our manuscript. EasySpin is only used for loading data in MATLAB.

**Reviewer 3**

The second harmonic detection scheme is presented as a means to distinguish between EPR signals of different conductive species in a graphite anode of a Li-ion battery. Some of the features of the investigated samples appear to be unexpected by the authors. As a consequence, EPR spectra were recorded with a modulation amplitude that was too large for some of the signal components, leading to overmodulation. To facilitate the separation of different signal contributions, second harmonic detection was employed. Thereby, a different contrast was obtained, favoring signals from metallic lithium. It would be interesting to also see the effect of varying the modulation amplitude. However, as discussed in previous comments, some of the experiments cannot be easily reproduced. Nonetheless, the use of second harmonic detection provides an additional, independent method for signal separation. This technique represents a welcome addition to the tool set available to EPR spectroscopists for the investigation of battery components and batteries. It should be reported to the magnetic resonance community, even though some questions regarding the investigated samples may remain unanswered at this point. The presented data are suitable and sufficient to substantiate the claims of the manuscript. As a basis for decision making for other researchers, a discussion of the signal amplitudes caused by the different sources (or, more precisely, their attenuation) would be helpful. (This comment is directly related to comment 1 of reviewer #2 in RC8.) Furthermore, the A/B ratio is used as a qualitative measure to identify the signal origin. Would a variation of A/B be expected with second harmonic detection, considering that the signal originates from multiple sources, or from species with a continuous distribution of relevant length scales? After these minor revisions, this reviewer recommends publication of the manuscript.

**Response:** We thank the reviewer for this very positive appraisal of the work. As suggested by the reviewer and the reviewer #2, we added a sentence concerning the EPR selectivity using the second harmonic detection mode and two corresponding bibliographic references (on page 6, line 121-122).
Concerning the A/B ratio, we thank the reviewer for his question. The variation of this ratio depends on the different $Li^0$ morphologies (micro-aggregates, mossy or dendritic structures). Usually, thick $Li^0$ exhibits an asymmetric lineshape (1st harmonic - Dysonian), mossy $Li^0$ a slightly asymmetric peak and dendritic $Li^0$ a pure Lorentzian line. Consequently, the second harmonic EPR signal of $Li^0$ micro-aggregates will exhibit an asymmetric lineshape while dendritic $Li^0$ structures will show a symmetric EPR spectrum, with an apparent intensity higher than micro-aggregates (S/N ratio higher). An example of such second harmonic lineshapes is given in following Figure.

[Figure]